# Style Transfer from Non-Parallel Text by Cross-Alignment

**Tianxiao Shen**[1]    **Tao Lei**[2]    **Regina Barzilay**[1]    **Tommi Jaakkola**[1]

[1]MIT CSAIL        [2]ASAPP Inc.

[1]{tianxiao, regina, tommi}@csail.mit.edu  [2]tao@asapp.com

## Abstract

This paper focuses on style transfer on the basis of non-parallel text. This is an instance of a broad family of problems including machine translation, decipherment, and sentiment modification. The key challenge is to separate the content from other aspects such as style. We assume a shared latent content distribution across different text corpora, and propose a method that leverages refined alignment of latent representations to perform style transfer. The transferred sentences from one style should match example sentences from the other style as a population. We demonstrate the effectiveness of this cross-alignment method on three tasks: sentiment modification, decipherment of word substitution ciphers, and recovery of word order.[1]

## 1 Introduction

Using massive amounts of parallel data has been essential for recent advances in text generation tasks, such as machine translation and summarization. However, in many text generation problems, we can only assume access to non-parallel or mono-lingual data. Problems such as decipherment or style transfer are all instances of this family of tasks. In all of these problems, we must preserve the content of the source sentence but render the sentence consistent with desired presentation constraints (e.g., style, plaintext/ciphertext).

The goal of controlling one aspect of a sentence such as style independently of its content requires that we can disentangle the two. However, these aspects interact in subtle ways in natural language sentences, and we can succeed in this task only approximately even in the case of parallel data. Our task is more challenging here. We merely assume access to two corpora of sentences with the same distribution of content albeit rendered in different styles. Our goal is to demonstrate that this distributional equivalence of content, if exploited carefully, suffices for us to learn to map a sentence in one style to a style-independent content vector and then decode it to a sentence with the same content but a different style.

In this paper, we introduce a refined alignment of sentence representations across text corpora. We learn an encoder that takes a sentence and its original style indicator as input, and maps it to a style-independent content representation. This is then passed to a style-dependent decoder for rendering. We do not use typical VAEs for this mapping since it is imperative to keep the latent content representation rich and unperturbed. Indeed, richer latent content representations are much harder to align across the corpora and therefore they offer more informative content constraints. Moreover, we reap additional information from cross-generated (style-transferred) sentences, thereby getting two distributional alignment constraints. For example, positive sentences that are style-transferred into negative sentences should match, as a population, the given set of negative sentences. We illustrate this cross-alignment in Figure 1.

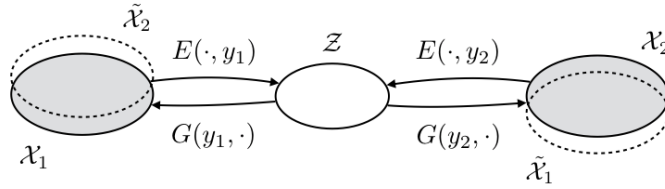

Figure 1: An overview of the proposed cross-alignment method. $\mathcal{X}_1$ and $\mathcal{X}_2$ are two sentence domains with different styles $y_1$ and $y_2$, and $\mathcal{Z}$ is the shared latent content space. Encoder $E$ maps a sentence to its content representation, and generator $G$ generates the sentence back when combining with the original style. When combining with a different style, transferred $\tilde{\mathcal{X}}_1$ is aligned with $\mathcal{X}_2$ and $\tilde{\mathcal{X}}_2$ is aligned with $\mathcal{X}_1$ at the distributional level.

To demonstrate the flexibility of the proposed model, we evaluate it on three tasks: sentiment modification, decipherment of word substitution ciphers, and recovery of word order. In all of these applications, the model is trained on non-parallel data. On the sentiment modification task, the model successfully transfers the sentiment while keeps the content for 41.5% of review sentences according to human evaluation, compared to 41.0% achieved by the control-gen model of Hu et al. (2017). It achieves strong performance on the decipherment and word order recovery tasks, reaching Bleu score of 57.4 and 26.1 respectively, obtaining 50.2 and 20.9 gap than a comparable method without cross-alignment.

## 2 Related work

**Style transfer in vision** Non-parallel style transfer has been extensively studied in computer vision (Gatys et al., 2016; Zhu et al., 2017; Liu and Tuzel, 2016; Liu et al., 2017; Taigman et al., 2016; Kim et al., 2017; Yi et al., 2017). Gatys et al. (2016) explicitly extract content and style features, and then synthesize a new image by combining "content" features of one image with "style" features from another. More recent approaches learn generative networks directly via generative adversarial training (Goodfellow et al., 2014) from two given data domains $\boldsymbol{X}_1$ and $\boldsymbol{X}_2$. The key computational challenge in this non-parallel setting is aligning the two domains. For example, CoupledGANs (Liu and Tuzel, 2016) employ weight-sharing between networks to learn cross-domain representation, whereas CycleGAN (Zhu et al., 2017) introduces cycle consistency which relies on transitivity to regularize the transfer functions. While our approach has a similar high-level architecture, the discreteness of natural language does not allow us to reuse these models and necessitates the development of new methods.

**Non-parallel transfer in natural language** In natural language processing, most tasks that involve generation (e.g., translation and summarization) are trained using parallel sentences. Our work most closely relates to approaches that do not utilize parallel data, but instead guide sentence generation from an indirect training signal (Mueller et al., 2017; Hu et al., 2017). For instance, Mueller et al. (2017) manipulate the hidden representation to generate sentences that satisfy a desired property (e.g., sentiment) as measured by a corresponding classifier. However, their model does not necessarily enforce content preservation. More similar to our work, Hu et al. (2017) aims at generating sentences with controllable attributes by learning disentangled latent representations (Chen et al., 2016). Their model builds on variational auto-encoders (VAEs) and uses independency constraints to enforce that attributes can be reliably inferred back from generated sentences. While our model builds on distributional cross-alignment for the purpose of style transfer and content preservation, these constraints can be added in the same way.

**Adversarial training over discrete samples** Recently, a wide range of techniques addresses challenges associated with adversarial training over discrete samples generated by recurrent networks (Yu et al., 2016; Lamb et al., 2016; Hjelm et al., 2017; Che et al., 2017). In our work, we employ the Professor-Forcing algorithm (Lamb et al., 2016) which was originally proposed to close the gap between teacher-forcing during training and self-feeding during testing for recurrent networks. This design fits well with our scenario of style transfer that calls for cross-alignment. By using

continuous relaxation to approximate the discrete sampling process (Jang et al., 2016; Maddison et al., 2016), the training procedure can be effectively optimized through back-propagation (Kusner and Hernández-Lobato, 2016; Goyal et al., 2017).

## 3 Formulation

In this section, we formalize the task of non-parallel style transfer and discuss the feasibility of the learning problem. We assume the data are generated by the following process:

1. a latent style variable $\boldsymbol{y}$ is generated from some distribution $p(\boldsymbol{y})$;
2. a latent content variable $\boldsymbol{z}$ is generated from some distribution $p(\boldsymbol{z})$;
3. a datapoint $\boldsymbol{x}$ is generated from conditional distribution $p(\boldsymbol{x}|\boldsymbol{y}, \boldsymbol{z})$.

We observe two datasets with the same content distribution but different styles $\boldsymbol{y}_1$ and $\boldsymbol{y}_2$, where $\boldsymbol{y}_1$ and $\boldsymbol{y}_2$ are unknown. Specifically, the two observed datasets $\boldsymbol{X}_1 = \{\boldsymbol{x}_1^{(1)}, \cdots, \boldsymbol{x}_1^{(n)}\}$ and $\boldsymbol{X}_2 = \{\boldsymbol{x}_2^{(1)}, \cdots, \boldsymbol{x}_2^{(m)}\}$ consist of samples drawn from $p(\boldsymbol{x}_1|\boldsymbol{y}_1)$ and $p(\boldsymbol{x}_2|\boldsymbol{y}_2)$ respectively. We want to estimate the style transfer functions between them, namely $p(\boldsymbol{x}_1|\boldsymbol{x}_2; \boldsymbol{y}_1, \boldsymbol{y}_2)$ and $p(\boldsymbol{x}_2|\boldsymbol{x}_1; \boldsymbol{y}_1, \boldsymbol{y}_2)$.

A question we must address is when this estimation problem is feasible. Essentially, we only observe the marginal distributions of $\boldsymbol{x}_1$ and $\boldsymbol{x}_2$, yet we are going to recover their joint distribution:

$$p(\boldsymbol{x}_1, \boldsymbol{x}_2|\boldsymbol{y}_1, \boldsymbol{y}_2) = \int_{\boldsymbol{z}} p(\boldsymbol{z}) p(\boldsymbol{x}_1|\boldsymbol{y}_1, \boldsymbol{z}) p(\boldsymbol{x}_2|\boldsymbol{y}_2, \boldsymbol{z}) d\boldsymbol{z} \tag{1}$$

As we only observe $p(\boldsymbol{x}_1|\boldsymbol{y}_1)$ and $p(\boldsymbol{x}_2|\boldsymbol{y}_2)$, $\boldsymbol{y}_1$ and $\boldsymbol{y}_2$ are unknown to us. If two different $\boldsymbol{y}$ and $\boldsymbol{y}'$ lead to the same distribution $p(\boldsymbol{x}|\boldsymbol{y}) = p(\boldsymbol{x}|\boldsymbol{y}')$, then given a dataset $\boldsymbol{X}$ sampled from it, its underlying style can be either $\boldsymbol{y}$ or $\boldsymbol{y}'$. Consider the following two cases: (1) both datasets $\boldsymbol{X}_1$ and $\boldsymbol{X}_2$ are sampled from the same style $\boldsymbol{y}$; (2) $\boldsymbol{X}_1$ and $\boldsymbol{X}_2$ are sampled from style $\boldsymbol{y}$ and $\boldsymbol{y}'$ respectively. These two scenarios have different joint distributions, but the observed marginal distributions are the same. To prevent such confusion, we constrain the underlying distributions as stated in the following proposition:

**Proposition 1.** *In the generative framework above, $\boldsymbol{x}_1$ and $\boldsymbol{x}_2$'s joint distribution can be recovered from their marginals only if for any different $\boldsymbol{y}, \boldsymbol{y}' \in \mathcal{Y}$, distributions $p(\boldsymbol{x}|\boldsymbol{y})$ and $p(\boldsymbol{x}|\boldsymbol{y}')$ are different.*

This proposition basically says that $\boldsymbol{X}$ generated from different styles should be "distinct" enough, otherwise the transfer task between styles is not well defined. While this seems trivial, it may not hold even for simplified data distributions. The following examples illustrate how the transfer (and recovery) becomes feasible or infeasible under different model assumptions. As we shall see, for a certain family of styles $\mathcal{Y}$, the more complex distribution for $\boldsymbol{z}$, the more probable it is to recover the transfer function and the easier it is to search for the transfer.

### 3.1 Example 1: Gaussian

Consider the common choice that $\boldsymbol{z} \sim \mathcal{N}(\boldsymbol{0}, \boldsymbol{I})$ has a centered isotropic Gaussian distribution. Suppose a style $\boldsymbol{y} = (\boldsymbol{A}, \boldsymbol{b})$ is an affine transformation, i.e. $\boldsymbol{x} = \boldsymbol{A}\boldsymbol{z} + \boldsymbol{b} + \boldsymbol{\epsilon}$, where $\boldsymbol{\epsilon}$ is a noise variable. For $\boldsymbol{b} = \boldsymbol{0}$ and any orthogonal matrix $\boldsymbol{A}$, $\boldsymbol{A}\boldsymbol{z} + \boldsymbol{b} \sim N(\boldsymbol{0}, \boldsymbol{I})$ and hence $\boldsymbol{x}$ has the same distribution for any such styles $\boldsymbol{y} = (\boldsymbol{A}, \boldsymbol{0})$. In this case, the effect of rotation cannot be recovered.

Interestingly, if $\boldsymbol{z}$ has **a more complex distribution**, such as a Gaussian mixture, then affine transformations can be uniquely determined.

**Lemma 1.** *Let $\boldsymbol{z}$ be a mixture of Gaussians $p(\boldsymbol{z}) = \sum_{k=1}^{K} \pi_k \mathcal{N}(\boldsymbol{z}; \boldsymbol{\mu}_k, \boldsymbol{\Sigma}_k)$. Assume $K \geq 2$, and there are two different $\boldsymbol{\Sigma}_i \neq \boldsymbol{\Sigma}_j$. Let $\mathcal{Y} = \{(\boldsymbol{A}, \boldsymbol{b}) || \boldsymbol{A}| \neq 0\}$ be all invertible affine transformations, and $p(\boldsymbol{x}|\boldsymbol{y}, \boldsymbol{z}) = \mathcal{N}(\boldsymbol{x}; \boldsymbol{A}\boldsymbol{z} + \boldsymbol{b}, \epsilon^2 \boldsymbol{I})$, in which $\epsilon$ is a noise. Then for all $\boldsymbol{y} \neq \boldsymbol{y}' \in \mathcal{Y}$, $p(\boldsymbol{x}|\boldsymbol{y})$ and $p(\boldsymbol{x}|\boldsymbol{y}')$ are different distributions.*

**Theorem 1.** *If the distribution of $\boldsymbol{z}$ is a mixture of Gaussians which has more than two different components, and $\boldsymbol{x}_1, \boldsymbol{x}_2$ are two affine transformations of $\boldsymbol{z}$, then the transfer between them can be recovered given their respective marginals.*

## 3.2 Example 2: Word substitution

Consider here another example when $z$ is a bi-gram language model and a style $y$ is a vocabulary in use that maps each "content word" onto its surface form (lexical form). If we observe two realizations $x_1$ and $x_2$ of the same language $z$, the transfer and recovery problem becomes inferring a word alignment between $x_1$ and $x_2$.

Note that this is a simplified version of language decipherment or translation. Nevertheless, the recovery problem is still sufficiently hard. To see this, let $M_1, M_2 \in \mathcal{R}^{n \times n}$ be the estimated bi-gram probability matrix of data $X_1$ and $X_2$ respectively. Seeking the word alignment is equivalent to finding a permutation matrix $P$ such that $P^\top M_1 P \approx M_2$, which can be expressed as an optimization problem,

$$\min_{P} \; \|P^\top M_1 P - M_2\|^2$$

The same formulation applies to graph isomorphism (GI) problems given $M_1$ and $M_2$ as the adjacency matrices of two graphs, suggesting that determining the existence and uniqueness of $P$ is at least GI hard. Fortunately, if $M$ as a graph is complex enough, the search problem could be more tractable. For instance, if each vertex's weights of incident edges as a set is unique, then finding the isomorphism can be done by simply matching the sets of edges. This assumption largely applies to our scenario where $z$ is a complex language model. We empirically demonstrate this in the results section.

The above examples suggest that $z$ as the latent content variable should carry most complexity of data $x$, while $y$ as the latent style variable should have relatively simple effects. We construct the model accordingly in the next section.

## 4 Method

Learning the style transfer function under our generative assumption is essentially learning the conditional distribution $p(x_1|x_2; y_1, y_2)$ and $p(x_2|x_1; y_1, y_2)$. Unlike in vision where images are continuous and hence the transfer functions can be learned and optimized directly, the discreteness of language requires us to operate through the latent space. Since $x_1$ and $x_2$ are conditionally independent given the latent content variable $z$,

$$
\begin{aligned}
p(x_1|x_2; y_1, y_2) &= \int_z p(x_1, z | x_2; y_1, y_2) dz \\
&= \int_z p(z|x_2, y_2) \cdot p(x_1|y_1, z) dz \\
&= \mathbb{E}_{z \sim p(z|x_2, y_2)}[p(x_1|y_1, z)]
\end{aligned}
\tag{2}
$$

This suggests us learning an auto-encoder model. Specifically, a style transfer from $x_2$ to $x_1$ involves two steps—an encoding step that infers $x_2$'s content $z \sim p(z|x_2, y_2)$, and a decoding step which generates the transferred counterpart from $p(x_1|y_1, z)$. In this work, we approximate and train $p(z|x, y)$ and $p(x|y, z)$ using neural networks (where $y \in \{y_1, y_2\}$).

Let $E : \mathcal{X} \times \mathcal{Y} \to \mathcal{Z}$ be an encoder that infers the content $z$ for a given sentence $x$ and a style $y$, and $G : \mathcal{Y} \times \mathcal{Z} \to \mathcal{X}$ be a generator that generates a sentence $x$ from a given style $y$ and content $z$. $E$ and $G$ form an auto-encoder when applying to the same style, and thus we have reconstruction loss,

$$
\begin{aligned}
\mathcal{L}_{\text{rec}}(\theta_E, \theta_G) = \; & \mathbb{E}_{x_1 \sim X_1}[-\log p_G(x_1|y_1, E(x_1, y_1))] + \\
& \mathbb{E}_{x_2 \sim X_2}[-\log p_G(x_2|y_2, E(x_2, y_2))]
\end{aligned}
\tag{3}
$$

where $\theta$ are the parameters to estimate.

In order to make a meaningful transfer by flipping the style, $X_1$ and $X_2$'s content space must coincide, as our generative framework presumed. To constrain that $x_1$ and $x_2$ are generated from the same latent content distribution $p(z)$, one option is to apply a variational auto-encoder (Kingma and Welling, 2013). A VAE imposes a prior density $p(z)$, such as $z \sim \mathcal{N}(0, I)$, and uses a KL-divergence regularizer to align both posteriors $p_E(z|x_1, y_1)$ and $p_E(z|x_2, y_2)$ to it,

$$
\mathcal{L}_{\text{KL}}(\theta_E) = \; \mathbb{E}_{x_1 \sim X_1}[D_{\text{KL}}(p_E(z|x_1, y_1)\|p(z))] + \mathbb{E}_{x_2 \sim X_2}[D_{\text{KL}}(p_E(z|x_2, y_2)\|p(z))]
\tag{4}
$$

The overall objective is to minimize $\mathcal{L}_{\text{rec}} + \mathcal{L}_{\text{KL}}$, whose opposite is the variational lower bound of data likelihood.

However, as we have argued in the previous section, restricting $\boldsymbol{z}$ to a simple and even distribution and pushing most complexity to the decoder may not be a good strategy for non-parallel style transfer. In contrast, a standard auto-encoder simply minimizes the reconstruction error, encouraging $\boldsymbol{z}$ to carry as much information about $\boldsymbol{x}$ as possible. On the other hand, it lowers the entropy in $p(\boldsymbol{x}|\boldsymbol{y},\boldsymbol{z})$, which helps to produce meaningful style transfer in practice as we flip between $\boldsymbol{y}_1$ and $\boldsymbol{y}_2$. Without explicitly modeling $p(\boldsymbol{z})$, it is still possible to force distributional alignment of $p(\boldsymbol{z}|\boldsymbol{y}_1)$ and $p(\boldsymbol{z}|\boldsymbol{y}_2)$. To this end, we introduce two constrained variants of auto-encoder.

## 4.1 Aligned auto-encoder

Dispense with VAEs that make an explicit assumption about $p(\boldsymbol{z})$ and align both posteriors to it, we align $p_E(\boldsymbol{z}|\boldsymbol{y}_1)$ and $p_E(\boldsymbol{z}|\boldsymbol{y}_2)$ with each other, which leads to the following constrained optimization problem:

$$\boldsymbol{\theta}^* = \arg\min_{\boldsymbol{\theta}} \mathcal{L}_{\text{rec}}(\boldsymbol{\theta}_E, \boldsymbol{\theta}_G)$$

$$\text{s.t.} \quad E(\boldsymbol{x}_1, \boldsymbol{y}_1) \overset{\text{d}}{=} E(\boldsymbol{x}_2, \boldsymbol{y}_2) \qquad \boldsymbol{x}_1 \sim \boldsymbol{X}_1, \boldsymbol{x}_2 \sim \boldsymbol{X}_2 \tag{5}$$

In practice, a Lagrangian relaxation of the primal problem is instead optimized. We introduce an adversarial discriminator $D$ to align the aggregated posterior distribution of $\boldsymbol{z}$ from different styles (Makhzani et al., 2015). $D$ aims to distinguish between these two distributions:

$$\mathcal{L}_{\text{adv}}(\boldsymbol{\theta}_E, \boldsymbol{\theta}_D) = \mathbb{E}_{\boldsymbol{x}_1 \sim \boldsymbol{X}_1}[-\log D(E(\boldsymbol{x}_1, \boldsymbol{y}_1))] + \mathbb{E}_{\boldsymbol{x}_2 \sim \boldsymbol{X}_2}[-\log(1 - D(E(\boldsymbol{x}_2, \boldsymbol{y}_2)))] \tag{6}$$

The overall training objective is a min-max game played among the encoder $E$, generator $G$ and discriminator $D$. They constitute an aligned auto-encoder:

$$\min_{E,G} \max_{D} \mathcal{L}_{\text{rec}} - \lambda \mathcal{L}_{\text{adv}} \tag{7}$$

We implement the encoder $E$ and generator $G$ using single-layer RNNs with GRU cell. $E$ takes an input sentence $\boldsymbol{x}$ with initial hidden state $\boldsymbol{y}$, and outputs the last hidden state $\boldsymbol{z}$ as its content representation. $G$ generates a sentence $\boldsymbol{x}$ conditioned on latent state $(\boldsymbol{y}, \boldsymbol{z})$. To align the distributions of $\boldsymbol{z}_1 = E(\boldsymbol{x}_1, \boldsymbol{y}_1)$ and $\boldsymbol{z}_2 = E(\boldsymbol{x}_2, \boldsymbol{y}_2)$, the discriminator $D$ is a feed-forward network with a single hidden layer and a sigmoid output layer.

## 4.2 Cross-aligned auto-encoder

The second variant, cross-aligned auto-encoder, directly aligns the transfered samples from one style with the true samples from the other. Under the generative assumption, $p(\boldsymbol{x}_2|\boldsymbol{y}_2) = \int_{\boldsymbol{x}_1} p(\boldsymbol{x}_2|\boldsymbol{x}_1; \boldsymbol{y}_1, \boldsymbol{y}_2) p(\boldsymbol{x}_1|\boldsymbol{y}_1) d\boldsymbol{x}_1$, thus $\boldsymbol{x}_2$ (sampled from the left-hand side) should exhibit the same distribution as transfered $\boldsymbol{x}_1$ (sampled from the right-hand side), and vice versa. Similar to our first model, the second model uses two discriminators $D_1$ and $D_2$ to align the populations. $D_1$'s job is to distinguish between real $\boldsymbol{x}_1$ and transfered $\boldsymbol{x}_2$, and $D_2$'s job is to distinguish between real $\boldsymbol{x}_2$ and transfered $\boldsymbol{x}_1$.

Adversarial training over the discrete samples generated by $G$ hinders gradients propagation. Although sampling-based gradient estimator such as REINFORCE (Williams, 1992) can by adopted, training with these methods can be unstable due to the high variance of the sampled gradient. Instead, we employ two recent techniques to approximate the discrete training (Hu et al., 2017; Lamb et al., 2016). First, instead of feeding a single sampled word as the input to the generator RNN, we use the softmax distribution over words instead. Specifically, during the generating process of transfered $\boldsymbol{x}_2$ from $G(\boldsymbol{y}_1, \boldsymbol{z}_2)$, suppose at time step $t$ the output logit vector is $\boldsymbol{v}_t$. We feed its peaked distribution softmax$(\boldsymbol{v}_t/\gamma)$ as the next input, where $\gamma \in (0,1)$ is a temperature parameter.

Secondly, we use Professor-Forcing (Lamb et al., 2016) to match the sequence of hidden states instead of the output words, which contains the information about outputs and is smoothly distributed. That is, the input to the discriminator $D_1$ is the sequence of hidden states of either (1) $G(\boldsymbol{y}_1, \boldsymbol{z}_1)$ teacher-forced by a real example $\boldsymbol{x}_1$, or (2) $G(\boldsymbol{y}_1, \boldsymbol{z}_2)$ self-fed by previous soft distributions.

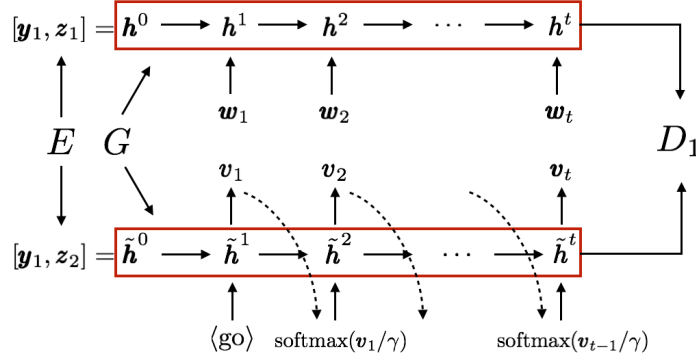

Figure 2: Cross-aligning between $\boldsymbol{x}_1$ and transferred $\boldsymbol{x}_2$. For $\boldsymbol{x}_1$, $G$ is teacher-forced by its words $\boldsymbol{w}_1\boldsymbol{w}_2\cdots\boldsymbol{w}_t$. For transfered $\boldsymbol{x}_2$, $G$ is self-fed by previous output logits. The sequence of hidden states $\boldsymbol{h}^0,\cdots,\boldsymbol{h}^t$ and $\tilde{\boldsymbol{h}}^0,\cdots,\tilde{\boldsymbol{h}}^t$ are passed to discriminator $D_1$ to be aligned. Note that our first variant aligned auto-encoder is a special case of this, where only $\boldsymbol{h}^0$ and $\tilde{\boldsymbol{h}}^0$, i.e. $\boldsymbol{z}_1$ and $\boldsymbol{z}_2$, are aligned.

---

**Algorithm 1** Cross-aligned auto-encoder training. The hyper-parameters are set as $\lambda = 1, \gamma = 0.001$ and learning rate is $0.0001$ for all experiments in this paper.

---

**Input:** Two corpora of different styles $\boldsymbol{X}_1, \boldsymbol{X}_2$. Lagrange multiplier $\lambda$, temperature $\gamma$.

Initialize $\boldsymbol{\theta}_E, \boldsymbol{\theta}_G, \boldsymbol{\theta}_{D_1}, \boldsymbol{\theta}_{D_2}$

**repeat**

    **for** $p = 1, 2; q = 2, 1$ **do**

        Sample a mini-batch of $k$ examples $\{\boldsymbol{x}_p^{(i)}\}_{i=1}^k$ from $\boldsymbol{X}_p$

        Get the latent content representations $\boldsymbol{z}_p^{(i)} = E(\boldsymbol{x}_p^{(i)}, \boldsymbol{y}_p)$

        Unroll $G$ from initial state $(\boldsymbol{y}_p, \boldsymbol{z}_p^{(i)})$ by feeding $\boldsymbol{x}_p^{(i)}$, and get the hidden states sequence $\boldsymbol{h}_p^{(i)}$

        Unroll $G$ from initial state $(\boldsymbol{y}_q, \boldsymbol{z}_p^{(i)})$ by feeding previous soft output distribution with temper-

        ature $\gamma$, and get the transferred hidden states sequence $\tilde{\boldsymbol{h}}_p^{(i)}$

    **end for**

    Compute the reconstruction $\mathcal{L}_{\text{rec}}$ by Eq. (3)

    Compute $D_1$'s (and symmetrically $D_2$'s) loss:

$$\mathcal{L}_{\text{adv}_1} = -\frac{1}{k}\sum_{i=1}^k \log D_1(\boldsymbol{h}_1^{(i)}) - \frac{1}{k}\sum_{i=1}^k \log(1 - D_1(\tilde{\boldsymbol{h}}_2^{(i)})) \tag{8}$$

    Update $\{\boldsymbol{\theta}_E, \boldsymbol{\theta}_G\}$ by gradient descent on loss

$$\mathcal{L}_{\text{rec}} - \lambda(\mathcal{L}_{\text{adv}_1} + \mathcal{L}_{\text{adv}_2}) \tag{9}$$

    Update $\boldsymbol{\theta}_{D_1}$ and $\boldsymbol{\theta}_{D_2}$ by gradient descent on loss $\mathcal{L}_{\text{adv}_1}$ and $\mathcal{L}_{\text{adv}_2}$ respectively

**until** convergence

**Output:** Style transfer functions $G(\boldsymbol{y}_2, E(\cdot, \boldsymbol{y}_1)) : \mathcal{X}_1 \to \mathcal{X}_2$ and $G(\boldsymbol{y}_1, E(\cdot, \boldsymbol{y}_2)) : \mathcal{X}_2 \to \mathcal{X}_1$

---

The running procedure of our cross-aligned auto-encoder is illustrated in Figure 2. Note that cross-aligning strengthens the alignment of latent variable $\boldsymbol{z}$ over the recurrent network of generator $G$. By aligning the whole sequence of hidden states, it prevents $\boldsymbol{z}_1$ and $\boldsymbol{z}_2$'s initial misalignment from propagating through the recurrent generating process, as a result of which the transferred sentence may end up somewhere far from the target domain.

We implement both $D_1$ and $D_2$ using convolutional neural networks for sequence classification (Kim, 2014). The training algorithm is presented in Algorithm 1.

# 5 Experimental setup

**Sentiment modification** Our first experiment focuses on text rewriting with the goal of changing the underlying sentiment, which can be regarded as "style transfer" between negative and positive sentences. We run experiments on Yelp restaurant reviews, utilizing readily available user ratings associated with each review. Following standard practice, reviews with rating above three are considered positive, and those below three are considered negative. While our model operates at the sentence level, the sentiment annotations in our dataset are provided at the document level. We assume that all the sentences in a document have the same sentiment. This is clearly an oversimplification, since some sentences (e.g., background) are sentiment neutral. Given that such sentences are more common in long reviews, we filter out reviews that exceed 10 sentences. We further filter the remaining sentences by eliminating those that exceed 15 words. The resulting dataset has 250K negative sentences, and 350K positive ones. The vocabulary size is 10K after replacing words occurring less than 5 times with the "<unk>" token. As a baseline model, we compare against the control-gen model of Hu et al. (2017).

To quantitatively evaluate the transfered sentences, we adopt a model-based evaluation metric similar to the one used for image transfer (Isola et al., 2016). Specifically, we measure how often a transferred sentence has the correct sentiment according to a pre-trained sentiment classifier. For this purpose, we use the TextCNN model as described in Kim (2014). On our simplified dataset for style transfer, it achieves nearly perfect accuracy of 97.4%.

While the quantitative evaluation provides some indication of transfer quality, it does not capture all the aspects of this generation task. Therefore, we also perform two human evaluations on 500 sentences randomly selected from the test set[2]. In the first evaluation, the judges were asked to rank generated sentences in terms of their fluency and sentiment. Fluency was rated from 1 (unreadable) to 4 (perfect), while sentiment categories were "positive", "negative", or "neither" (which could be contradictory, neutral or nonsensical). In the second evaluation, we evaluate the transfer process comparatively. The annotator was shown a source sentence and the corresponding outputs of the systems in a random order, and was asked "Which transferred sentence is semantically equivalent to the source sentence with an opposite sentiment?". They can be both satisfactory, A/B is better, or both unsatisfactory. We collect two labels for each question. The label agreement and conflict resolution strategy can be found in the supplementary material. Note that the two evaluations are not redundant. For instance, a system that always generates the same grammatically correct sentence with the right sentiment independently of the source sentence will score high in the first evaluation setup, but low in the second one.

**Word substitution decipherment** Our second set of experiments involves decipherment of word substitution ciphers, which has been previously explored in NLP literature (Dou and Knight, 2012; Nuhn and Ney, 2013). These ciphers replace every word in plaintext (natural language) with a cipher token according to a 1-to-1 substitution key. The decipherment task is to recover the plaintext from ciphertext. It is trivial if we have access to parallel data. However we are interested to consider a non-parallel decipherment scenario. For training, we select 200K sentences as $X_1$, and apply a substitution cipher $f$ on a different set of 200K sentences to get $X_2$. While these sentences are non-parallel, they are drawn from the same distribution from the review dataset. The development and test sets have 100K parallel sentences $D_1 = \{x^{(1)}, \cdots, x^{(n)}\}$ and $D_2 = \{f(x^{(1)}), \cdots, f(x^{(n)})\}$. We can quantitatively compare between $D_1$ and transferred (deciphered) $D_2$ using Bleu score (Papineni et al., 2002).

Clearly, the difficulty of this decipherment task depends on the number of substituted words. Therefore, we report model performance with respect to the percentage of the substituted vocabulary. Note that the transfer models do not know that $f$ is a word substitution function. They learn it entirely from the data distribution.

In addition to having different transfer models, we introduce a simple decipherment baseline based on word frequency. Specifically, we assume that words shared between $X_1$ and $X_2$ do not require translation. The rest of the words are mapped based on their frequency, and ties are broken arbitrarily. Finally, to assess the difficulty of the task, we report the accuracy of a machine translation system trained on a parallel corpus (Klein et al., 2017).

| Method | accuracy |
|---|---|
| Hu et al. (2017) | 83.5 |
| Variational auto-encoder | 23.2 |
| Aligned auto-encoder | 48.3 |
| Cross-aligned auto-encoder | 78.4 |

Table 1: Sentiment accuracy of transferred sentences, as measured by a pretrained classifier.

| Method | sentiment | fluency | overall transfer |
|---|---|---|---|
| Hu et al. (2017) | 70.8 | 3.2 | 41.0 |
| Cross-align | 62.6 | 2.8 | 41.5 |

Table 2: Human evaluations on sentiment, fluency and overall transfer quality. Fluency rating is from 1 (unreadable) to 4 (perfect). Overall transfer quality is evaluated in a comparative manner, where the judge is shown a source sentence and two transferred sentences, and decides whether they are both good, both bad, or one is better.

**Word order recovery**   Our final experiments focus on the word ordering task, also known as bag translation (Brown et al., 1990; Schmaltz et al., 2016). By learning the style transfer functions between original English sentences $X_1$ and shuffled English sentences $X_2$, the model can be used to recover the original word order of a shuffled sentence (or conversely to randomly permute a sentence). The process to construct non-parallel training data and parallel testing data is the same as in the word substitution decipherment experiment. Again the transfer models do not know that $f$ is a shuffle function and learn it completely from data.

## 6   Results

**Sentiment modification**   Table 1 and Table 2 show the performance of various models for both human and automatic evaluation. The control-gen model of Hu et al. (2017) performs better in terms of sentiment accuracy in both evaluations. This is not surprising because their generation is directly guided by a sentiment classifier. Their system also achieves higher fluency score. However, these gains do not translate into improvements in terms of the overall transfer, where our model faired better. As can be seen from the examples listed in Table 3, our model is more consistent with the grammatical structure and semantic meaning of the source sentence. In contrast, their model achieves sentiment change by generating an entirely new sentence which has little overlap with the original. The discrepancy between the two experiments demonstrate the crucial importance of developing appropriate evaluation measures for comparing methods for style transfer.

**Word substitution decipherment**   Table 4 summarizes the performance of our model and the baselines on the decipherment task, at various levels of word substitution. Consistent with our intuition, the last row in this table shows that the task is trivial when the parallel data is provided. In non-parallel case, the difficulty of the task is driven by the substitution rate. Across all the testing conditions, our cross-aligned model consistently outperforms its counterparts. The difference becomes more pronounced as the task becomes harder. When the substitution rate is 20%, all methods do a reasonably good job in recovering substitutions. However, when 100% of the words are substituted (as expected in real language decipherment), the poor performance of variational autoencoder and aligned auto-encoder rules out their application for this task.

**Word order recovery**   The last column in Table 4 demonstrates the performance on the word order recovery task. Order recovery is much harder—even when trained with parallel data, the machine translation model achieves only 64.6 Bleu score. Note that some generated orderings may be completely valid (e.g., reordering conjunctions), but the models will be penalized for producing them. In this task, only the cross-aligned auto-encoder achieves grammatical reorder to a certain extent, demonstrated by its Bleu score 26.1. Other models fail this task, doing no better than no transfer.

| | From negative to positive |
| --- | --- |
| | consistently slow . |
| | consistently good . |
| | consistently fast . |
| | |
| | my goodness it was so gross . |
| | my husband 's steak was phenomenal . |
| | my goodness was so awesome . |
| | |
| | it was super dry and had a weird taste to the entire slice . |
| | it was a great meal and the tacos were very kind of good . |
| | it was super flavorful and had a nice texture of the whole side . |

| | From positive to negative |
| --- | --- |
| | i love the ladies here ! |
| | i avoid all the time ! |
| | i hate the doctor here ! |
| | |
| | my appetizer was also very good and unique . |
| | my bf was n't too pleased with the beans . |
| | my appetizer was also very cold and not fresh whatsoever . |
| | |
| | came here with my wife and her grandmother ! |
| | came here with my wife and hated her ! |
| | came here with my wife and her son . |

Table 3: Sentiment transfer samples. The first line is an input sentence, the second and third lines are the generated sentences after sentiment transfer by Hu et al. (2017) and our cross-aligned auto-encoder, respectively.

| Method | Substitution decipher | | | | | Order recover |
| --- | --- | --- | --- | --- | --- | --- |
| | 20% | 40% | 60% | 80% | 100% | |
| No transfer (copy) | 56.4 | 21.4 | 6.3 | 4.5 | 0 | 5.1 |
| Unigram matching | 74.3 | 48.1 | 17.8 | 10.7 | 1.2 | - |
| Variational auto-encoder | 79.8 | 59.6 | 44.6 | 34.4 | 0.9 | 5.3 |
| Aligned auto-encoder | 81.0 | 68.9 | 50.7 | 45.6 | 7.2 | 5.2 |
| Cross-aligned auto-encoder | **83.8** | **79.1** | **74.7** | **66.1** | **57.4** | **26.1** |
| Parallel translation | 99.0 | 98.9 | 98.2 | 98.5 | 97.2 | 64.6 |

Table 4: Bleu scores of word substitution decipherment and word order recovery.

## 7 Conclusion

Transferring languages from one style to another has been previously trained using parallel data. In this work, we formulate the task as *a decipherment problem* with access only to non-parallel data. The two data collections are assumed to be generated by a latent variable generative model. Through this view, our method optimizes neural networks by forcing distributional alignment (invariance) over the latent space or sentence populations. We demonstrate the effectiveness of our method on tasks that permit quantitative evaluation, such as sentiment transfer, word substitution decipherment and word ordering. The decipherment view also provides an interesting open question—*when can the joint distribution $p(x_1, x_2)$ be recovered given only marginal distributions?* We believe addressing this general question would promote the style transfer research in both vision and NLP.

## Acknowledgments

We thank Nicholas Matthews for helping to facilitate human evaluations, and Zhiting Hu for sharing his code. We also thank Jonas Mueller, Arjun Majumdar, Olga Simek, Danelle Shah, MIT NLP group and the reviewers for their helpful comments. This work was supported by MIT Lincoln Laboratory.

## Footnotes

[1]Our code and data are available at https://github.com/shentianxiao/language-style-transfer.

[2]we eliminated 37 sentences from them that were judged as neutral by human judges.

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

## A   Proof of Lemma 1

**Lemma 1.** *Let $z$ be a mixture of Gaussians $p(z) = \sum_{k=1}^{K} \pi_k \mathcal{N}(z; \boldsymbol{\mu}_k, \boldsymbol{\Sigma}_k)$. Assume $K \geq 2$, and there are two different $\boldsymbol{\Sigma}_i \neq \boldsymbol{\Sigma}_j$. Let $\mathcal{Y} = \{(\boldsymbol{A}, \boldsymbol{b}) || \boldsymbol{A} | \neq 0\}$ be all invertible affine transformations, and $p(\boldsymbol{x}|\boldsymbol{y}, \boldsymbol{z}) = \mathcal{N}(\boldsymbol{x}; \boldsymbol{A}\boldsymbol{z} + \boldsymbol{b}, \epsilon^2 \boldsymbol{I})$, in which $\epsilon$ is a noise. Then for all $\boldsymbol{y} \neq \boldsymbol{y}' \in \mathcal{Y}$, $p(\boldsymbol{x}|\boldsymbol{y})$ and $p(\boldsymbol{x}|\boldsymbol{y}')$ are different distributions.*

*Proof.*

$$p(\boldsymbol{x}|\boldsymbol{y} = (\boldsymbol{A}, \boldsymbol{b})) = \sum_{k=1}^{K} \pi_k \mathcal{N}(\boldsymbol{x}; \boldsymbol{A}\boldsymbol{\mu}_k + \boldsymbol{b}, \boldsymbol{A}\boldsymbol{\Sigma}_k \boldsymbol{A}^\top + \epsilon^2 \boldsymbol{I})$$

For different $\boldsymbol{y} = (\boldsymbol{A}, \boldsymbol{b})$ and $\boldsymbol{y}' = (\boldsymbol{A}', \boldsymbol{b}')$, $p(\boldsymbol{x}|\boldsymbol{y}) = p(\boldsymbol{x}|\boldsymbol{y}')$ entails that for $k = 1, \cdots, K$,

$$\begin{cases} \boldsymbol{A}\boldsymbol{\mu}_k + \boldsymbol{b} = \boldsymbol{A}'\boldsymbol{\mu}_k + \boldsymbol{b}' \\ \boldsymbol{A}\boldsymbol{\Sigma}_k \boldsymbol{A}^\top = \boldsymbol{A}'\boldsymbol{\Sigma}_k \boldsymbol{A}'^\top \end{cases}$$

Since all $\mathcal{Y}$ are invertible,

$$(\boldsymbol{A}^{-1}\boldsymbol{A}')\boldsymbol{\Sigma}_k (\boldsymbol{A}^{-1}\boldsymbol{A}')^\top = \boldsymbol{\Sigma}_k$$

Suppose $\boldsymbol{\Sigma}_k = \boldsymbol{Q}_k \boldsymbol{D}_k \boldsymbol{Q}_k^\top$ is $\boldsymbol{\Sigma}_k$'s orthogonal diagonalization. If $k = 1$, all solutions for $\boldsymbol{A}^{-1}\boldsymbol{A}'$ have the form:

$$\left\{ \boldsymbol{Q}\boldsymbol{D}^{1/2}\boldsymbol{U}\boldsymbol{D}^{-1/2}\boldsymbol{Q}^\top \middle| \boldsymbol{U} \text{ is orthogonal} \right\}$$

However, when $K \geq 2$ and there are two different $\boldsymbol{\Sigma}_i \neq \boldsymbol{\Sigma}_j$, the only solution is $\boldsymbol{A}^{-1}\boldsymbol{A}' = \boldsymbol{I}$, i.e. $\boldsymbol{A} = \boldsymbol{A}'$, and thus $\boldsymbol{b} = \boldsymbol{b}'$.

Therefore, for all $\boldsymbol{y} \neq \boldsymbol{y}'$, $p(\boldsymbol{x}|\boldsymbol{y}) \neq p(\boldsymbol{x}|\boldsymbol{y}')$. $\qquad\square$

