[Reviews · NeurIPS 2017]

Reviewer 1



This paper develops a method for performing style transfer between two linguistic domains using only un-paired examples. The authors separate out style and content within a probabilistic adversarial auto-encoder framework, implemented as RNN encoders/generators with feed-forward (convolutional) discriminators. Evaluations on sentiment reversal, word substition decipherment and word order recovery experiments show that the methods outperform simple baselines as well as a stronger VAE baseline, but still fall short of training on actual parallel text (as is to be expected). This is a good paper. The modifications are well-motivated and the model exposition is very clearly presented. The experimental results seem believable and show that the methods clearly outperform variational auto-encoders. Overall I think this work represents a good step in the right direction for linguistic style transfer research. Nit: Choose one spelling for {cypher, cipher}.

Reviewer 2



This paper proposes a method for style transfer between natural language sentences that does not require any actual examples of style-to-style translation, only unmatched sets of examples in each style. This builds on the high-level idea of the Zhu et al. CycleGAN, but is technically quite different to accommodate the different requirements of discrete sequence generation. The paper is clear, includes some clever applications of an adversarial objective, and I see no serious technical issues. The resulting style-transferred sentences are short and a bit odd, but the task is ambitious enough that even this strikes me as a substantial accomplishment, and the analytic results suggest that this approach to modeling is basically on the right track. The paper's comparison with the Hu et al. Controllable Text Generation paper, which attempts to solve roughly the same problem, is a bit weak. The two papers are chronologically close enough that I don't think this should be grounds for rejection, but it could be done better. The VAE baseline in this paper is never fully described, so it's hard to tell how well it can be used as a proxy to understand how well the Hu et al. model would do. If that baseline is not closely comparable to Hu et al.'s model, it would be helpful to include their as a baseline. 26-34: The initial description of the model here took me a while to understand, and didn't completely make sense until I'd seen more of the paper. I suspect a visual depiction of the model at this point could help quite a bit. Figure 1 could be clearer. The lines without arrowheads are a bit confusing.

Reviewer 3



This paper presents a method for learning style transfer models based on non-parallel corpora. The premise of the work is that it is possible to disentangle the style from the content and that when there are two different corpora on the same content but in distinctly different styles, then it is possible to induce the content and the style components. While part of me is somewhat skeptical whether it is truly possible to separate out the style from the content of natural language text, and that I tend to think sentiment and word-reordering presented in this work as applications correspond more to the content of an article than the style, I do believe that this paper presents a very creative and interesting exploration that makes both theoretical and empirical contributions. — (relatively minor) questions: - In the aligned auto-encoder setting, the discriminator is based on a simple feedforward NN, while in the cross-aligned auto-encoder setting, the discriminators are based on convolution NNs. I imagine ConvNets make stronger discriminators, thus it’d be helpful if the paper can shed lights on how much the quality of the discriminators influence the overall performance of the generators. - Some details on the implementation and experiments are missing, which might matter for reproducibility. For example, what kind of RNNs are used for the encoder and the generator? What kind of convolution networks are used for the discriminators? What are the vocabulary sizes (and the corresponding UNK scheme) for different datasets? The description for the word reordering dataset is missing. If the authors will share their code and the dataset, it’d help resolving most these questions, but still it’d be good to have some of these detailed specified in the final manuscript.